# Experimental Research of Abnormal Wear for Water-Lubricated Polymer Bearings under Low Speed, Heavy Pressure, and High Water Temperature

**DOI:** 10.3390/polym15051227

**Published:** 2023-02-28

**Authors:** Ying Liu, Gengyuan Gao, Dan Jiang

**Affiliations:** 1School of Mechanical Engineering, Shanghai Jiao Tong University, Shanghai 200240, China; 2School of Design, Shanghai Jiao Tong University, Shanghai 200240, China

**Keywords:** water lubricated, polymer bearings, operating conditions, hydrolysis failure

## Abstract

Polymer bearings used in a real ship had a hydrolysis failure under 50 rpm at 0.5 MPa with 40 °C water temperature. The test conditions were determined based on the operating conditions of the real ship. The test equipment was rebuilt to accommodate bearing sizes in a real ship. Water swelling was eliminated after 6 months’ soaking. The results showed that the polymer bearing was subjected to hydrolysis because of the increased heat generation and heat dissipation deterioration under low speed, heavy pressure, and high water temperature. The wear depth in the hydrolysis area is 10 times larger than that in normal wear area, and the melting, stripping, transferring, adhering, and accumulation of hydrolyzed polymers caused abnormal wear. Additionally, extensive cracking was observed in the hydrolysis area of the polymer bearing.

## 1. Introduction

Stern tube bearings are an important part of a ship’s power system. It has become an irreversible trend to replace oil lubrication with water lubrication because of its environmentally friendly properties. Water-lubricated stern tube bearings are generally made of non-metallic materials, among which polymer bearings are the most commonly used. The operating conditions of polymer bearings generally include low speed and heavy pressure, and even operating under water temperatures of nearly 40 °C in summer. Whether the polymer bearing can withstand such harsh operating conditions is crucial. Therefore, it is necessary to study the performance of polymer bearings under low speed, heavy pressure, and high water temperature.

Some studies involved the effect of water flow on the bush temperature of water-lubricated bearings [1,2,3]. The closed lubrication system was used, and the water flow could be controlled artificially [4]. Temperature sensors were buried to measure bearing surface temperature, which is under the bearing and several millimeters away from the bearing’s worn surface [5]. Litwin [6,7,8] used different water flows to lubricate the test bearing. The result showed abnormal wear failure of polymer bearings without water flow lubrication after operating for a couple of minutes [9,10]. Litwin also [11,12] studied the thermal effect of water-lubricated stern tube bearings. The result showed that the intensity of forced axial flow has a strong influence on bush temperature.

Agnieszka Barszczewska [13,14] studied polymer bearings operating with insufficient water lubrication, and the result showed that the polymer bearing would undergo hydrolysis and melt after operating for 60 min without water flowing. The influence of load and water flow on bearing temperature was considered. Meanwhile, the test error of temperature sensors installed at both ends of the polymer bearing was also considered. Some other studies were devoted to carrying capacity [15,16,17] and performance of polymer bearings under mixed lubrication conditions [18,19,20,21], axis misalignment [22,23,24,25,26], thermally induced seizure failure [27], thermal effects on a hydrodynamic bearing’s clearance [28], and different operation conditions [29].

The effect of water flow on bearing temperature is critical, especially for polymer bearings. However, the speeds, specific pressures, and water temperature also have a crucial influence on the operation of polymer bearings. Furthermore, some extreme operating conditions of low speeds, heavy pressures, and high temperatures can also cause rapid increases in bearing surface temperatures, eventually leading to melting and the hydrolysis of polymers. Among them, polymer bearings are most sensitive to the changing of water temperature, because water temperature has the greatest influence on the heat dissipation conditions of test bearings. When three unfavorable conditions of low speed, heavy pressure, and high temperature exist simultaneously, the bearings are prone to abnormal hydrolysis failure. Hydrolytic failure refers to the softening and chemical reaction of the bearing material when the polymer bearing is operated under high temperature and harsh lubrication conditions, resulting in a change in material properties [13,14]. According to the observation of experimental phenomena [7], hydrolysis occurs in a short period of time and leads to abnormal failure of test bearings, significantly reducing their service life.

In the actual operation of polymer bearings in real ships, an open lubrication system is generally adopted to replace closed lubrication systems [10]. Stern bearings are completely soaked in water for open lubrication systems in real ships, and, for polymer bearings, the diameter will decrease because of water absorption. Hence, the test bearing must be soaked in water for several months to eliminate water swelling on the diameter. In this article, the operating conditions in real ships are mainly considered. In order to make the test conditions consistent with the operating conditions in real ships, an open seawater lubrication system has been adopted. Polymer bearings used in real ships are used as the test bearings. The operating conditions are low speeds, heavy pressures, and high water temperature, which are set according to real ship operating conditions.

## 2. Experimental Equipment and Operating Conditions

### 2.1. Experimental Equipment and Data Measurement

The schematic diagram of the test equipment is shown in Figure 1, which consists of a loading system, performance test system, operating control system, and mechanical system. The loading system mainly applies load to the polymer bearing, adopting hydraulic loading. Figure 2 shows the diagram of the loading device, where pressure is applied to test the bearings by controlling the up and down operation of the stern shaft by hydraulic pressure. The performance test system consists of a temperature sensor, force sensor, and torque sensor [6,7]. The temperature sensors are installed at the lower end of the test bearing with an accuracy of 0.1 °C, which are located at the front, middle, and end of the bearing, shown in Figure 1. Because of the bearing wear during the test, temperature sensors measure the temperature at a distance of 3 mm from the inner surface of the polymer bearing [5]. The force sensors are installed at both ends of polymer bearing with accuracy of 1 N. The applied specific pressure needs to be converted into positive force by division by the force area [7]; in the manuscript, the force area is the product of the diameter and bearing length. The torque sensor is installed in the control cabinet with an accuracy of 1 N·m. The operating control system is mainly used to adjust the operating speeds, force value, and water temperatures, which is centrally arranged in the control cabinet.

The mechanical system consists of test polymer bearings, the stern shaft, and its fasteners. The stern shaft is made of a ZCuSn_10_Z_2_ alloy. The operating conditions of the polymer bearings in real ships can be simulated and controlled by the test equipment. Table 1 shows the properties of the polymer bearing bush under water lubrication. The L/D ratio of the test polymer bearing is 1.5, and the relative clearance value is 0.3%. To avoid shaft misalignment [28], when the stern and test bearing are installed, the plug gauge is used to measure the clearance between the shaft and test bearing on the left and right on the same horizontal plane. If the clearance between the left and right ends on the same plane is the same, the bearing axis and the stern axis are basically the same. To avoid stern shaft misalignment caused by propeller rotation, the stern shaft and propeller are separated, and propeller is mounted at the rear of the water tank, as shown in Figure 1.

The friction coefficient is calculated by torque, and the calculation formula is as follows:
$$f=\frac{T_{f}}{F*R}$$
where $T_{f}$ is the torque measured by the torque sensor and F is the normal force measured by the force sensor. R is the radius of the stern shaft as a definite value. The wear amount is the diameter difference before and after the test. The bearing is disassembled to measure the inner diameter with an inside diameter micrometer.


### 2.2. Operating Conditions

Figure 3 shows the test equipment in operation. The performance of the polymer bearing is tested under different speeds, specific pressures, and water temperatures. Since the load is equal to the product of the specific pressure and the force area, the specific pressure value is used instead of the load. Firstly, the performance of the polymer bearing under different speeds is tested, with speeds of 50, 100, 150, 200, 250, 300, and 350 rpm at the specific pressure of 0.2 MPa under the water temperature of 35 °C. Secondly, the speed and specific pressure are coupled together in the test; the test-specific pressures are 0.1, 0.2, 0.3, 0.4, and 0.5 MPa at every speed of 50–350 rpm, and the water temperature is also 35 °C. Thirdly, speed, specific pressure, and water temperature are coupled together in the test: the water temperatures are 20, 25, 30, 35, and 40 °C at speeds of 50–350 rpm, at normal operating pressure of 0.2 MPa and maximum permissible pressure of 0.5 MPa. During the test, the speed ranges first from high to low and then from low to high, eliminating the influence of speed changing the test result.

### 2.3. Preparation before Experiment Test

Before the test, the friction coefficient (COF) of the test equipment itself needs to be calibrated. The calibration tooling is shown in Figure 4; the specific pressures of 0.1–0.5 MPa and speeds of 50–350 rpm are all applied to the calibration tooling, calculating the COF under different speeds and pressures. The COF dependence of the test equipment is large with the specific pressures but remains weak with the rotational speed. When the specific pressure is ≤0.2 MPa, the COF is so small that the torque sensor cannot measure the value, and this part is neglected. When the specific pressure is at 0.3–0.4 MPa, the COF of the test equipment is about 0.0015 because of the torque sensor value range being small at 0.3–0.4 MPa. When the specific pressure is at 0.5 MPa, the friction coefficient of the test equipment is about 0.0030 because the torque sensor value is nearly doubled compared with that at 0.3–0.4 MPa. Thus, when calculating the bearing COF, it is necessary to subtract the torque value of the test equipment under different specific pressures, and then calculate the test bearing COF according to the formula in Section 2.1.

The water swelling of the polymer bearing is 1.3% in Table 1. Before the test, the polymer bearing needs to be soaked in water to eliminate water swelling. Water swelling causes the reduction of the polymer bearing’s diameter, further influencing the bearing performance and diameter wear amount. Figure 5b shows the picture of the polymer bearing before and after soaking. The polymer bearing diameter is tested before and after soaking for 6 months. Figure 5a shows the diameter measurement points inside the bearing, the A, B, and C sections in the front, middle, and end of the polymer bearing are selected, and 5 points are selected in each section; thus, there are a total of 15 diameter measurement points in the polymer bearing. The measuring points’ positions are marked in the non-carrying area, who’s diameter corresponds to the load carrying area. Among them, the diameter position of point 3 is just at the bottom of the bearing’s carrying area.

The diameters before and after soaking are shown in Table 2. Figure 6 shows the diameter reduction result before and after soaking. It is seen that point 3 has the lowest water swelling; meanwhile, points 1 and 5 have the highest water swelling, and points 3 and 4 have the middle water swelling at the A, B, and C sections. Because the polymer bearing is divided into two semicircular segments with possible installation gaps at the semicircular borders, the gap increases the water swelling at the edge points but has few influences on the middle point on each cross-section. As a result, points 1 and 5 have the highest water swelling and point 3 has the lowest water swelling. According to the 1.3% water swelling rate, based on the bush thickness of the polymer bearing, after 6 months’ soaking, the influence of water absorption on diameter is basically eliminated.

## 3. Results and Discussion

### 3.1. Friction Coefficient (COF) under Different Speeds

The lubrication state can be roughly judged by the change rate of the COF with increasing speeds. If the COF reduces faster with increasing speeds, it means that there are many actual contact points between the polymer bearing and stern shaft. If the COF hardly changes with increasing speeds, it means that the dynamic lubrication state is established. Figure 7 shows the COF under 50–350 rpm speeds at a specific pressure of 0.2 MPa with a water temperature of 35 °C; the COF decreases first and then tends to be stable with increasing speeds. At 50 rpm, the polymer bearing is in a mixed lubrication state [20]. Mixed lubrication combines the direct contact of asperities with fluid film [21]. The mixed lubrication characteristic is the comprehensive performance of various lubricating films. The proportion of various lubricating films on the friction pair is related to the operating speeds. The lubrication state is the worst at 50 rpm with more actual contact points between the bearing and stern shaft. At that moment, the surface roughness has an important influence on the COF. Thus, the highest COF occurs at 50 rpm.

At 100 rpm, the water-lubricated bearing is in an excessively lubricated state from mixed lubrication to hydrodynamic lubrication [27,28], which is better than the lubrication state at 50 rpm. The actual contact points in the friction pair are gradually filled with lubricating water. At that moment, the surface roughness has a lightened influence on the COF; hence, the COF reduces obviously at 100 rpm compared with that at 50 rpm. At 150–350 rpm, the hydrodynamic lubrication state is basically established, and, at that moment, the load carrying capacity of the water film makes the bearing no longer be in contact with the shaft; hence, surface roughness has no influence on the COF, and, as a result, the COF tends to be stable with increasing speeds. Additionally, at 150–350 rpm, the load-carrying capacity of the water film increases with increasing speeds. The COF has little decrease because of the increasing load-carrying capacity of the water film. Speeding up and speeding down have a limited effect on the COF.

### 3.2. Friction Coefficient (COF) under Different Speeds and Pressures

Figure 8a,b show the COF vs. speed under 0.1–0.5 MPa at a speed of 50–350 rpm. At every pressure of 0.1–0.5 MPa, the trends of the COF with speed are indeed consistent with that in Figure 7. Under 0.1–0.5 MPa, speed is also the main factor that affects the lubrication state at 50–350 rpm. Speeding up and speeding down have some effect on the COF, but the effect is also limited. Figure 8c shows the COF vs. pressure at certain speeds. In Figure 8c, at 50 rpm, the COF decreases with increasing pressures. At 100–350 rpm, the COF has little increase with increasing pressures.

At 50 rpm, the lubrication state is mixed lubrication, and the proportion of various lubricating films of the friction pair is various under different pressures [25]. The actual connect points increase with increasing pressures under mixed lubrication; otherwise, the roughness peak is rolled by heavy pressure, and, after that, the surface roughness of the connect points decreases continuously, causing the COF to decrease at 50 rpm with increasing pressures. Because speed is the main factor contributing to the lubrication state of polymer bearings, at 100–350 rpm, the lubrication state changes from mixed lubrication to hydrodynamic lubrication with increasing speeds. However, the water film becomes thin with specific pressure increasing from 0.1 MPa to 0.5 MPa, and, as a result, the lubrication state tends towards the mixed lubrication to form hydrodynamic lubrication. The surface roughness gradually affects the COF with increasing pressure; thus, the COF tends to increase with increasing pressure at 100–350 rpm.

### 3.3. Friction Coefficient (COF) under Different Speeds, Pressures, and Water Temperatures

Figure 9 shows the COF at different speeds, pressures, and water temperatures, and Figure 9a,c,e show the test result at a normal pressure of 0.2 MPa. Figure 9b,d,f show the test result at the maximum permissible pressure of 0.5 MPa. In Figure 9a,c,e, it is seen that speed is still the main factor affecting the COF at 20–40 °C under 0.2 MPa, and the COF decreases with increasing speeds. At a water temperature of 35 °C, the COF is significantly lower than that under other water temperatures at 0.2 MPa. Water temperature is nearly 35 °C in nature in August. After the tests of different speeds and pressures in Section 3.1 and Section 3.2, the polymer bearing has a certain adaptability to a water temperature of 35 °C at 0.2 MPa, so that the COF is the lowest at the water temperature of 35 °C at 0.2 MPa compared with that at other water temperatures.

In Figure 9b,d,f, it can be seen that at 0.5 MPa, water temperature has an improvement influence on the COF compared with that at 0.2 MPa. However, the test conditions include a low speed of 50 rpm, heavy pressure of 0.5 MPa, and high temperature of 40 °C, which are the most severe operating conditions for polymer bearings in real ships. The bearing surface produces more heat and dissipates less heat under the friction test at this time, resulting in a rapid rise in temperature of the polymer bearing surface; in a short time, the hydrolysis failure of the polymer bearing occurred [13].

Under the combination of low speed, heavy pressure, and high temperature, the lubrication state is close to dry friction and boundary lubrication. A large amount of friction heat is generated though the friction of the increasing actual connection points. What is worse, at a high temperature of 40 °C, the heat dissipation ability of lubrication water decreases obviously, and water amounts involved in lubrication also reduce obviously, causing the obvious weakening of heat dissipation conditions [14]. Finally, at 0.5 MPa with an increasing water temperature, the temperature rise between the water temperature and bearing surface temperature continues to increase [7], as shown in Figure 9g. Meanwhile, at normal operating pressures of 0.2 MPa, the temperature rise is stable with increasing water temperatures. Under 50 rpm at 0.5 MPa with water at 40 °C, the friction pair has the worst lubricating film, causing the most weakened heat dissipation ability. Heat accumulation causes the surface temperature of polymer bearings to rise rapidly, eventually leading to bearing hydrolytic failure in a short time.

### 3.4. Hydrolysis, Melting, and Polymer Accumulation of Polymer Bearing Surface

When the bearing was operated at 0.5 MPa, 50 rpm, and at 40 °C, some debris were observed in water tank surface, and then, a larger sheet or bulk of polymers gradually came out of the water. However, the COF still remained stable; thus, the test was continued. The polymer bearing was disassembled after the test. Figure 10 shows the pictures of the polymer bearing after the test; it is seen that the middle position of the polymer bearing was hydrolyzed in the whole ring, as shown in Figure 10a,b. The polymers were melted, stripped, transferred, and accumulated after hydrolysis; hence, the grooves in the middle position were blocked by stripped polymers, as shown in Figure 10c,d. The polymer bearing’s friction surface at the hydrolyzed part was covered with turtle cracks, as shown in Figure 10d. The polymers softened under high surface temperature, which then stripped and adhered to the surface of the polymer bearing and stern shaft during operation, as shown in Figure 10c,f [13,14]. This indicates that polymer bearings are sensitive to water temperature changes [7]. Especially under the combination of low speed, heavy pressure, and high temperature, abnormal melting hydrolysis failure occurred. The service life of polymer bearings is greatly shorted because of abnormal melting hydrolysis failure.

Figure 11 shows the diameter wear depth after polymer hydrolysis. The diameter of section B is largely increased because of the melting, stripping, and transferring of polymers after hydrolysis, and the hydrolysis depth in the diameter is about 1.5 mm in the B section. Meanwhile, the diameter wear depth of the A and C sections is 0.15 mm, because the A and C sections dissipated heat easily at the edge, so the hydrolysis and melting of polymers was not observed. Thus, the A and C sections have a normal diameter depth of 0.15 mm, one-tenth of the wear depth in section B. The polymer bearing at section B is abnormally worn because of hydrolysis [7].

Through the phenomenon observed under 0.5 MPa at 50 rpm and 40 °C, the bearing hydrolysis occurred in a short time. At that moment, much wear debris suddenly appeared on the water surface, and, after that, flakes and lumps of polymer appeared on the water surface gradually. However, during the process, the COF had no obvious range. The melting and hydrolysis of polymers is mainly caused by the increase in heat production and untimely heat dissipation under low speeds, heavy pressures, and high water temperatures in section B.

Figure 12 shows the average diameter change in section B during the whole test. Before the test, the water-lubricated bearing was soaked for 2 months, and the diameter gradually decreased because of water swelling. After being installed on the test equipment, the polymer bearings were constantly exposed to air waiting for the installation of other devices. After 4 months exposed, the bearing was disassembled to measure diameter, and it was found that the water swelling disappeared. Thus, water swelling needs to be eliminated again by continued soaking. Then, the polymer bearing was soaked for nearly 6 months again before the test, and the diameter decreased nearly 0.6 mm. After several diameter measurements, the diameter did not change significantly.

After the tests under different speeds, pressures, and water temperatures, the polymer bearing was melted and hydrolyzed suddenly, and then the bearing was disassembled. Finally, the diameter in section B increased 1.5 mm after hydrolysis under 50 rpm, 0.5 MPa, and 40 °C water temperature. Section B had abnormal wear because of insufficient heat dissipation and increasing heat production, and sections A and C had normal wear of 0.15 mm without hydrolysis and melting. Therefore, polymers are sensitive to drastic changes in water temperature from 20 °C to 40 °C. The combination of adverse factors of low speed, heavy pressure, and high temperature will produce the edge effect in the critical state, that is, at that moment, the bearing will undergo hydrolysis failure in a short time.

## 4. Conclusions

This article mainly studied polymer bearing performance under different speeds, pressures, and water temperatures. Meanwhile, the influence of water swelling on bearing diameter was studied. Finally, bearing hydrolysis, melting, striping, transferring, and adhering on the bearing surface was observed after 0.5 MPa, 50 rpm, and 40 °C water temperature tests. The hydrolysis causes abnormal wear, and the reasons for hydrolysis were analyzed. The main conclusions are as follows:Under different specific pressures, speeds, and water temperatures, speeds have the greatest impact on the lubrication state under 50–350 rpm; thus, speeds have great influence on the COF compared with specific pressures and water temperatures because the lubrication state affect the COF directly.The rising water temperature causes heat dissipation deterioration and obvious heat generation; thus, the temperature rise in the polymer bearing surface increases significantly at the maximum permissible specific pressure of 0.5 MPa. The hydrolysis and melting of the polymer bearing occurred under 50 rpm, 0.5 MPa, and a 40 °C water temperature.Water swelling time of polymer bearings is about 6 months, and bearing diameter after water swelling was reduced by about 0.6 mm. The wear depth after hydrolysis and melting was 10 times larger than normal wear depth. Polymer bearings are sensitive to high water temperature, heavy pressure, and low speed, and their occurrence at the same time needs to be avoided.

## Figures and Tables

**Figure 1 polymers-15-01227-f001:**
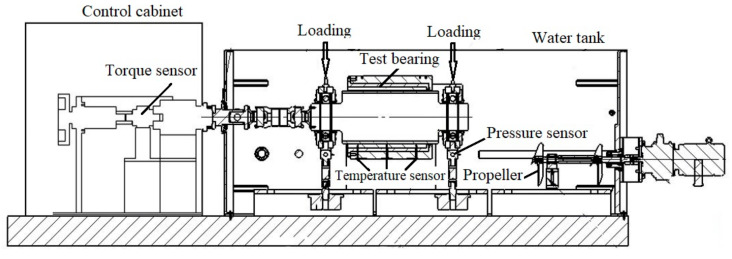
Schematic Diagram of Test Equipment.

**Figure 2 polymers-15-01227-f002:**
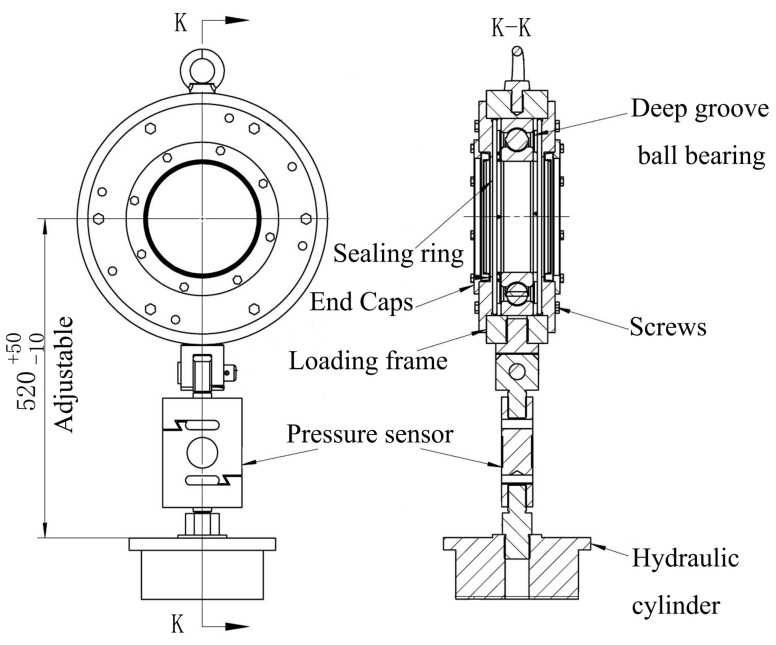
Diagram of the loading device.

**Figure 3 polymers-15-01227-f003:**
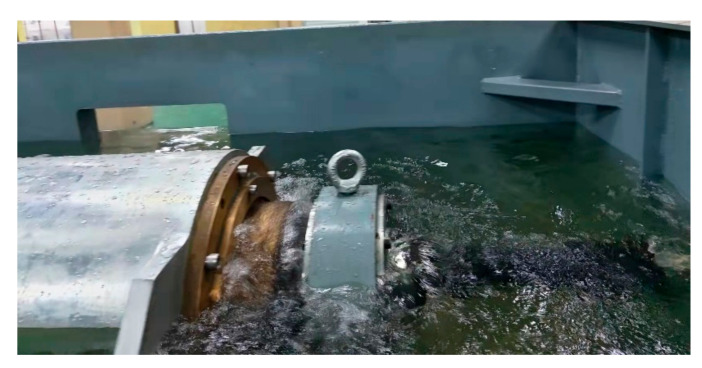
Pictures of test equipment in operation.

**Figure 4 polymers-15-01227-f004:**
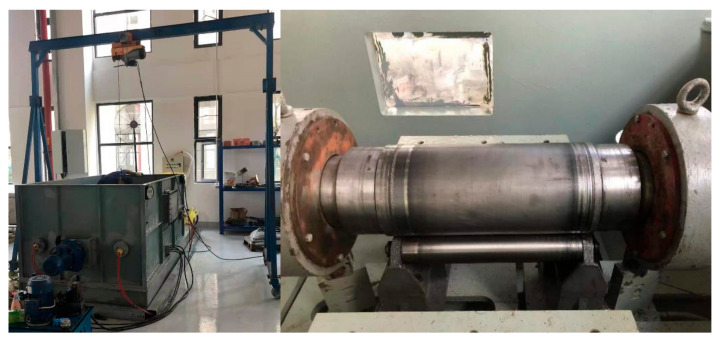
Calibration tooling of COF for test equipment.

**Figure 5 polymers-15-01227-f005:**
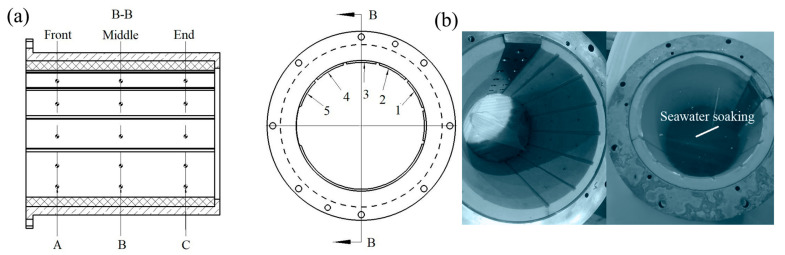
(**a**) Measuring point position of bearing diameter, (**b**) bearing pictures before and after soaking.

**Figure 6 polymers-15-01227-f006:**
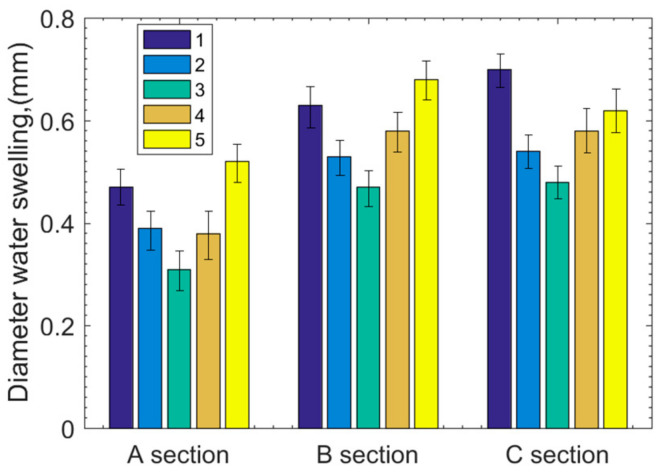
Diameter reduction before and after soaking for 6 months.

**Figure 7 polymers-15-01227-f007:**
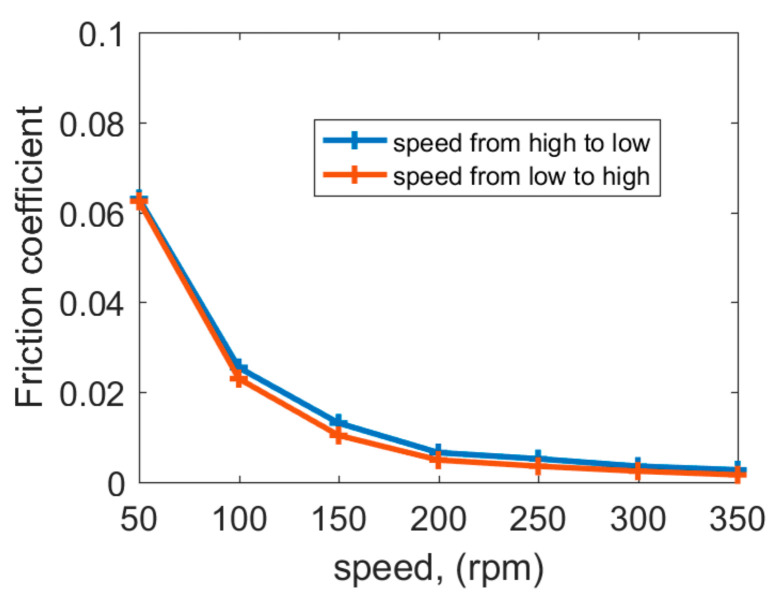
Friction coefficient vs. speeds at specific pressure of 0.2 MPa.

**Figure 8 polymers-15-01227-f008:**
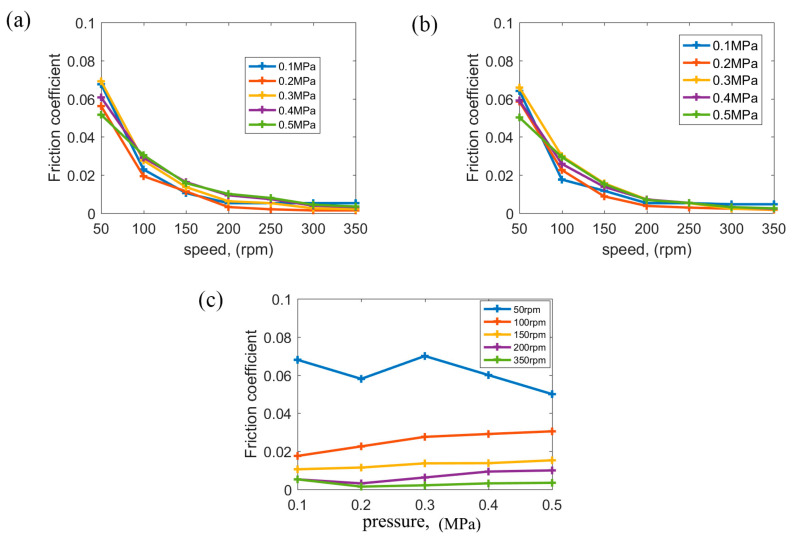
(**a**) COF vs. speeds with speeds ranging from high to low, (**b**) COF vs. speeds with speeds ranging from low to high, and (**c**) COF vs. pressures.

**Figure 9 polymers-15-01227-f009:**
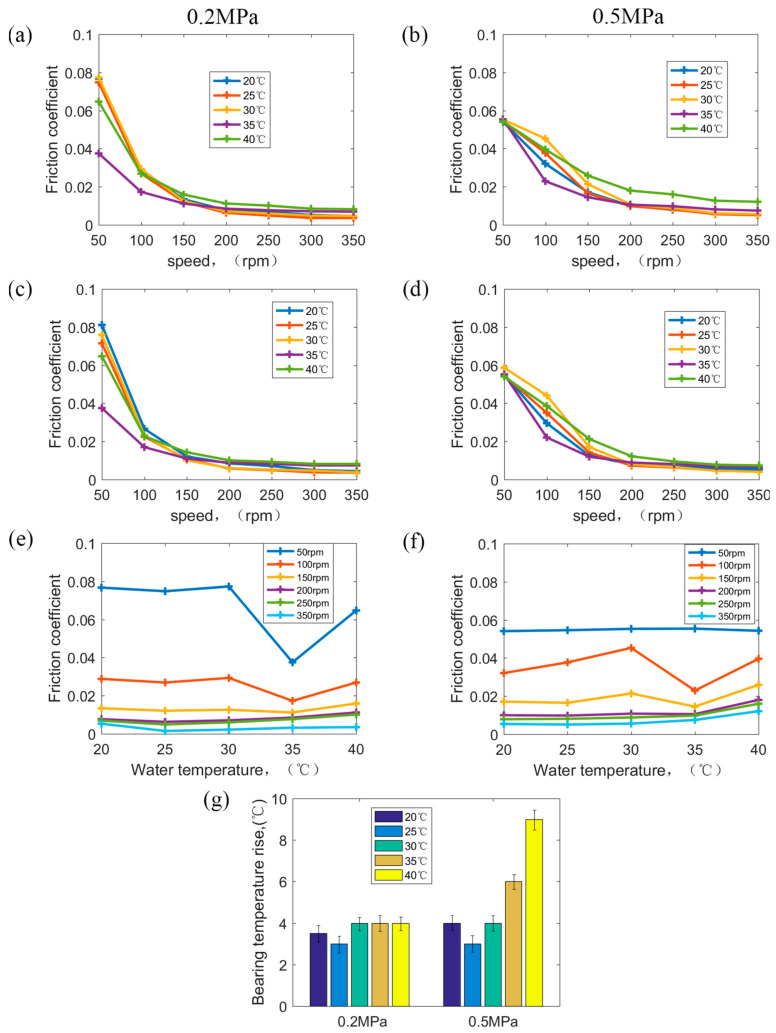
(**a**) COF vs. speeds at 0.2 MPa and (**b**) COF vs. speeds at 0.5 MPa with speeding down, (**c**) COF vs. speeds at 0.2 MPa and (**d**) COF vs. speeds at 0.5 MPa with speeding up. (**e**) COF vs. water temperature at 0.2 MPa, (**f**) COF vs. water temperature at 0.5 MPa. (**g**) Bearing temperature rise at different water temperatures under 0.2 MPa and 0.5 MPa.

**Figure 10 polymers-15-01227-f010:**
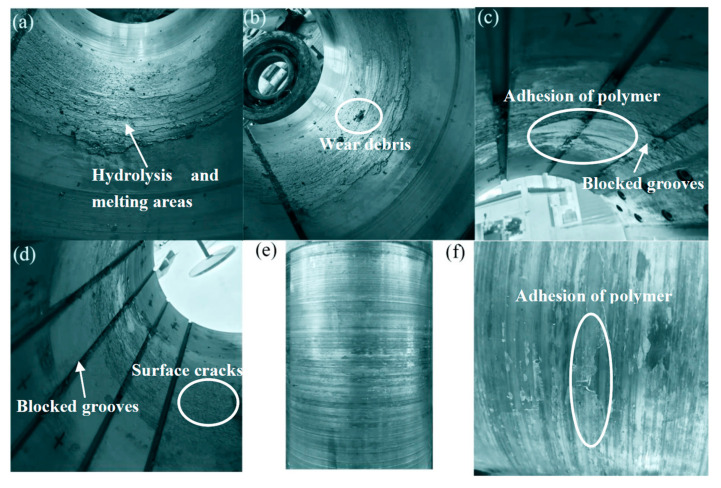
Picture of bearing surface and stern shaft surface after hydrolysis and melting. (**a**,**b**) Bearing loading area, (**c**,**d**) the blocked bearing grooves in non-load area. (**e**) Stern shaft surface and (**f**) enlarged view of (**e**).

**Figure 11 polymers-15-01227-f011:**
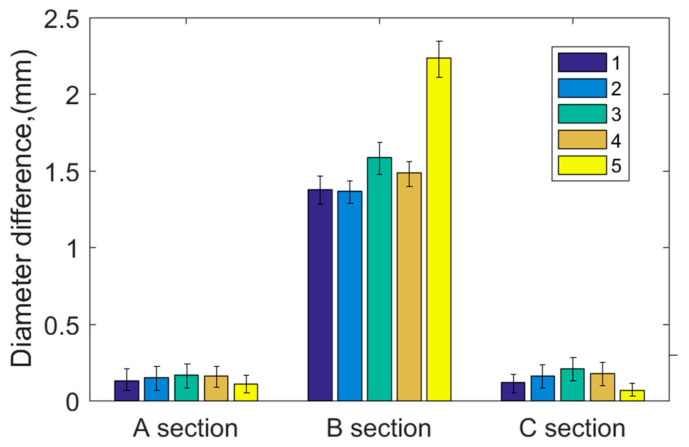
Diameter increase before and after hydrolysis.

**Figure 12 polymers-15-01227-f012:**
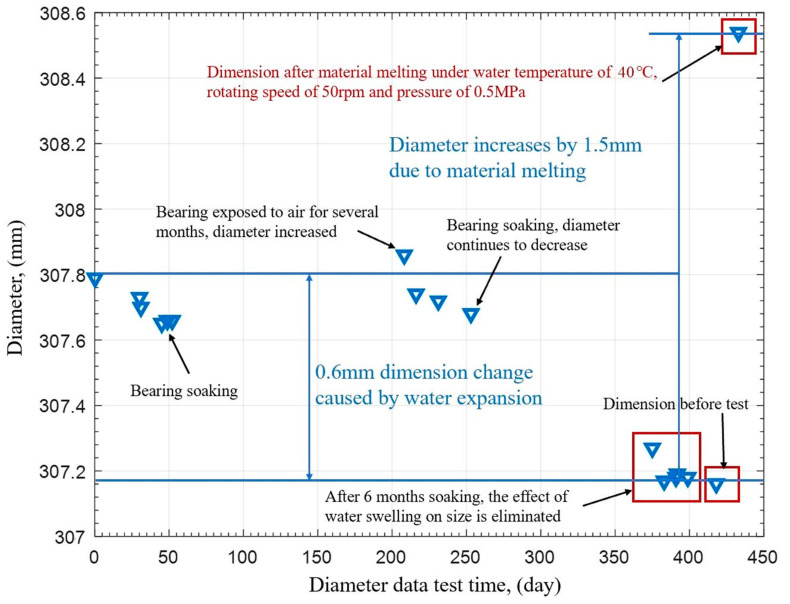
Changing of bearing diameter during whole test.

**Table 1 polymers-15-01227-t001:** Properties of polymer bearing bush.

Length/Diameter	Relative Clearance	Operating Temperature (wet)	Water Swell	Young’s Modulus	Poisson Ratio	Specific Gravity
1.5	0.3%	−7~60 °C	1.3%	605 MPa	~0.45	1.16

**Table 2 polymers-15-01227-t002:** Bearing diameter before and after soaking.

States	Section	Diameter of Measuring Points				
		1	2	3	4	5
Before soaking	A	307.92	307.95	307.92	307.90	307.90
	B	307.79	307.81	307.78	307.79	307.76
	C	307.70	307.69	307.63	307.63	307.60
After soaking	A	307.45	307.56	307.61	307.52	307.38
	B	307.16	307.28	307.31	307.21	307.08
	C	307.00	307.15	307.15	307.05	306.98

## Data Availability

Related research data can be obtained by contacting the author, Liu, Y.

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
