# Peer review of "Experimental Research of Abnormal Wear for Water-Lubricated Polymer Bearings under Low Speed, Heavy Pressure, and High Water Temperature"

_polymers, 2023, doi:10.3390/polym15051227_

Round 1
Reviewer 1 Report
This is an interesting experimental research work. Some improvement is needed before considering the manuscript for publication: the reviewer’s comments are the following:
1) The measured torque is not only due to the tested journal bearing: how is estimated or measured the parasite friction on rolling element bearings of the loading system, on the seals on the tank and on the shaft coupling (mechanical components located between the torque sensor and the tested journal bearing (polymer stern bearing))????
2) What is the accuracy of the torque measurement? What is the precision the deduced friction coefficient?
3) Usually, the diameter diminution due to the wear is not regular all around the circumference: how the authors improve the accuracy of the wear measurement?
4) Add a “s” at the word “sensor” in “Figure 2. Picture of pressure, torque, and temperature sensors.”
5) To the point of view of the reviewer and based on the test rig drawing (Fig. 1), it seems that the “pressure” sensor seems to be a “force” sensor, doesn’t it? If not, explain how the normal force Fn is deduced from the “pressure” measurement!
6) “… 50rpm, 100rpm, 150rpm, 200rpm, 250rpm, 300rpm, and 350rpm …” should be written “… 50, 100, 150, 200, 250, 300, and 350 rpm …” (note that a space is needed between the value “350” and the unit “rpm”).
7) In general, a space is needed between the value “…” and the unit “…” (for MPa, for example: 0.2Mpa should be 0.2 MPa.), except for the temperature: 40℃ is the correct writing.
8) “… 0.1MPa, 0.2MPa, 0.3MPa, 0.4MPa, and 0.5MPa …” should be written “… 0.1, 0.2, 0.3, 0.4 and 0.5 MPa …”
9) “… 50-350rpm…” should be written “… 50 - 350 rpm …”
10) “… 20℃, 25℃, 30℃, 35℃, and 40℃ …” should be written “… 20, 25, 30, 35 and 40℃ …”
11) “… the test load is 0.1, 0.2, 0.3, 0.4 and 0.5 MPa …” is not correct: either it is written “… the test specific pressure is 0.1, 0.2, 0.3, 0.4 and 0.5 MPa …” or “the test load” can be used but the values should be given in “N” or “kN”.
12) The authors assumed the friction coefficient of the test rig equipment is constant and equal to 0.0015. Isi t true whatever the radial load, the rotational speed and the water temperature?
13) On Figure 7 or in the caption of this figure, the applied load (or specific pressure) and the water temperature should be specified. This comment can be applied for other figures if some information is also missing.
14) Lines 162 & 163, it is written “Direction of speed change has limited effect on COF.”: this is not the “direction of speed”, but speed increasing (speeding-up) and speed decreasing (speeding-down).
15) Same comment for the writing at lines 170 & 171 “Speed direction has some effect on COF, but the effect is also limited.” : this is not the “direction of speed”, but speed increasing (speeding-up) and speed decreasing (speeding-down).
16) Lines 199 & 200, “In Figure 9 (b), (d), and (f), it seen, at 0.5MPa, that the water temperature has an improving influence on the COF compared to that at 0.2 MPa.” ….. should be written ….. “In Figure 9 (b), (d), and (f), it can be seen that, at 0.5MPa, the water temperature has an improving influence on the COF compared to that at 0.2 MPa.” or ….. “In Figure 9 (b), (d), and (f), it is seen, at 0.5MPa, that the water temperature has an improving influence on the COF compared to that at 0.2 MPa.”
17) The reviewer recommends to better explain and to define the definition of the hydrolysis phenomenon and hydrolysis wear (failure) at the beginning of the manuscript.
Author Response
Thank you for your review of the paper and for your objective and accurate comments. We have added our answers to the review comments in the attachment.

Reviewer 2 Report
The subject of the manuscript is certainly of great relevance: water lubrication of stern tubes became a widely used option for large tonnage vessels; polymers represent one of the solutions for bearing liners, hence the authors intended to evaluate experimentally their behaviour (namely, their friction coefficient and wear rate) in extreme conditions (low speeds, heavy loads and high temperatures).
However, the manuscript does not meet the minimum requirements to be accepted as a technical paper. The authors can find below some of the main points of criticisms.
1) The experiments are not clearly presented:
- the testing facility description lacks important information, such as the sizes of the bearing (the L/D ratio is a key parameter), the instrumentation (position of the thermocouples and load cells, their accuracy). Instead, one can find images of well-known sensors, which are not necessary relevant to the body of work. Figure 3 is also irrelevant. Questions such as “How is force applied?” or “Where is temperature measured?” (for such large sizes the temperature variation inside the bearing may be important) should be answered.
- How much is the clearance? testing a journal bearing without a control of the clearance is useless.
- There is no detail about the polymer material used; the variety of polymers is so large and their behaviour differs substantially; these results are useless without a material description.
2) Operating conditions are also poorly presented. Load expressed as a pressure must be explained. The formula at the top of page 3 uses unusual notations for mechanical engineers: Ff is traditionally used for friction force, not torque, and FN is defined as a positive pressure, which is not correct (it should be normal force).
3) Regarding the operating conditions, this reviewer has an important remark: the misalignment of propeller shafts is inevitable for long shafts, being thus a phenomenon that is present on the test-rig, as well. Is there any possibility to measure the shaft misalignment?
4) When analysing the experimental results, it is not clear how do the authors define the lubrication regime from the values of COF.
The list of references includes relevant papers related to the subject of water lubricated stern-tubes; however, it also includes some papers which are irrelevant, references [26] (seals with hard materials), [27] (non-conformal EHD contacts) and [28] being only a part of them. On the other hand, some papers addressing the phenomenon of hydrolysis of polymers could be also analysed and included within the bibliography.
Regarding the sizes of the test-rig, this reviewer has a remark: tests should not be carried out at such large dimensions in order to study wear; a smaller testing facility, in which contact pressures, speeds, and temperatures are reproduced, would be cheaper and easier to handle, while yielding to the same results. On the other hand, if the real operating conditions of the stern tube are intended to be simulated on this test-rig, it must be noted that the "propeller shaft" is subject to large flexural deformations; hence, the aligned shaft operation is not realistic.
As a conclusion, a great amount of the results presented in the manuscript are useless as long as the paper lacks key information, such as the materials tested, details of sizes, and operating conditions of the bearing. A comparison with similar results presented in those papers cited in the introduction is also recommendable or, alternately, some comparisons with theoretically predicted results; without these mentions, the experimental results are nothing but observations that are valid uniquely for the testing conditions. Such results could not be extrapolated to "real ship operating".
The text is written in a poor English; one can find typos and grammar errors in almost each paragraph. The paper must be carefully re-written in English. A few examples:
p.1,r10 “The results showed that the polymer bearing was hydrolysis because of heat production increase and heat dissipation deterioration “ - suggested formulation: “The results showed that the polymer bearing was subjected to hydrolysis because of the increased heat generation and heat dissipation deterioration”.
p.1,r38 " ...studied polymer bearing operating at insufficient water lubrication, the result showed that the would hydrolysis and melting after... " - there are 2 sentences here and “hydrolysis” is not a verb, while “melting” is a gerund verb, used incorrectly within the sentence.
p2, r46 " Although the effect of water flow on bearing temperature is critical, especially for 46 polymer bearing " the sentence is not finished.
p2, r63 " The operating conditions is tested under low speeds, heavy loads, and high-water temperature, which is set according to real ship operating " = the operating conditions ARE not tested; they are speed, load, temperature ... the sentence must be rephrased.”
p2, r68 " which is consists of " - “which consists of”
and so on....
Author Response
Thank you for your review of the paper and for your objective and accurate comments. We have added the responses to the reviewers' comments in the attachment.

Round 2
Reviewer 1 Report
The authors have considered the reviewer’s comments and recommendation and so, the revised version of this manuscript is more acceptable; however, few minor comments can still be done:
1) The English writing still needs to be improved.
2) “Force sensor is installed at both ends of the test bearing respectively with accuracy of 1 N. The applied pressure needs to be converted into positive force by multiplying with the force area [7], in the manuscript, the force area is the product of diameter and bearing length” ….. should be ….. “Force sensor is installed at both ends of the test bearing respectively with accuracy of 1 N. The applied pressure needs to be converted into positive force by dividing by the force area [7], in the manuscript, the force area is the product of diameter and bearing length”
3) “The COF of test equipment is large in relation to the specific pressures but little relation to speeds.” ….. should be …..“The COF dependence of test equipment is large with the specific pressure but remains weak with the rotational speed.”
Author Response
Thank you for your objective and accurate comments. We have added our answers to the review comments in the attachment.

Reviewer 2 Report
page 6 r. 176 ”the lubrcation regime is named HYDROdynamic (HD) lubrication”. Here I have a comment: In full film (HD) lubrication, friction torque varies with speed (see classical Petrov equation); hence COF will increase slowly because the increase in temperature counterbalance this effect by slightly reducing whilst in dry/boundary/mixed lubrication conditions friction. Moreover, the differences in Fig 7 are very small, within the acuracy of the measurements. This way of lubication regime definition is questionable.
Also here: " The mixed lubrication is a combination of dry friction, boundary lubrication, and hydrodynamic lubrication [21]" - To my knowledge, water has no ability to create boundary layers. Mixed lubrication combines direct contact of asperities with fluid film.
Correction in Table 2. Young's Modulus
Author Response

(The authors gave the same response as above.)
